# High-Efficiency Separation of Mg^2+^/Sr^2+^ through a NF Membrane under Electric Field

**DOI:** 10.3390/membranes12010057

**Published:** 2021-12-31

**Authors:** Huan Liu, Quan Li, Benqiao He, Zhengguang Sun, Feng Yan, Zhenyu Cui, Jianxin Li

**Affiliations:** 1State Key Laboratory of Separation Membranes and Membrane Processes, School of Materials Science and Engineering, Tiangong University, Tianjin 300387, China; lh962854@163.com (H.L.); leequan92@163.com (Q.L.); cuizheyhh@163.com (Z.C.); jxli@tiangong.edu.cn (J.L.); 2School of Materials Science & Engineering, Hubei University, Wuhan 430062, China; sunshine@hubu.edu.cn; 3School of Environmental Chemistry and Engineering, Tiangong University, Tianjin 300387, China; yanfeng@tiangong.edu.cn

**Keywords:** ENF process, congener ions, dehydration, efficient separation

## Abstract

The efficient separation of Sr^2+^/Mg^2+^ through nanofiltration (NF) technology is a great challenge because Sr^2+^ and Mg^2+^ ions are congeners with the same valence and chemical properties. In this work, an NF membrane under an electric field (EF) was successfully employed to separate Mg^2+^ and Sr^2+^ ions for the first time. The effects of current densities, Mg^2+^/Sr^2+^ mass ratios, pH of the feed, and coexisting cations on separation performance were investigated. Dehydration of Sr^2+^ or Mg^2+^ ions under EF was proved by molecular dynamics simulation. The results showed that a high-efficient separation of Mg^2+^/Sr^2+^ was achieved: Mg^2+^ removal of above 99% and increase in Sr^2+^ permeation with increasing EF. A separation factor reached 928 under optimal conditions, far higher than that without EF. The efficient separation of Mg^2+^/Sr^2+^ ions was mainly due to rejection of most Mg^2+^ ions by NF membrane and in situ precipitation of partly permeated Mg^2+^ ions by OH^−^ generated on the cathode under EF. Meanwhile, preferential dehydration of Sr^2+^ ions under EF due to lower hydration energy of Sr^2+^ compared with Mg^2+^ resulted in an increase of permeation of Sr^2+^ ions. This work provided a new idea for separation of congener ions with similar valence and chemical properties.

## 1. Introduction

Strontium (Sr) is known as “metal monosodium glutamate” for the development of high-tech products, such as semiconductor substrates, electronics, medicines and magnetic materials [1,2]. One of the Sr mineral resources is liquid brine in nature, such as saline lake brine, underground old brine and oil and gas field brine, the contents of which vary between 0.007 to 100 g/L [3,4,5]. Sr^2+^ ions often coexist with other cations in the brines, such as Na^+^, Li^+^, Ca^2+^ and Mg^2+^, which significantly increase separation difficulty and process cost. Conventional purification methods, such as chemical precipitation [6], solvent extraction [7,8], and adsorption [9,10], were used for Sr^2+^ extraction. However, these methods were often batch-mode and had a low separation efficiency, a high dose of additional reagents, and a risk of secondary environmental pollution due to wastewater discharge. Worse still, there remains a great challenge in separating the congener ions of Sr^2+^ and Mg^2+^ due to the similar physical and chemical properties. 

The membrane process, as a green technology, can provide continuous separation for purifying salts without adding precipitants and regenerants. The nanofiltration (NF) process exhibited high perm-selectivity especially for mono- and divalent ions largely depending on the repulsion effect for divalent ions, including size sieving, electrostatic repulsion, and dielectric repulsion [11,12]. As examples, Cheng et al. [13] used polyelectrolyte membranes (PEMs) to realize a certain separation of Sr^2+^ and Na^+^. Nicod et al. [14] reported that Sr^2+^ was separated from high concentration Na^+^ solution by NF. Increasing the size of Sr^2+^ species through complexation with polymers (PAA [15] or EI [16]) before membrane filtration realized the differentiation of Sr^2+^ ions from other coexisting ions. Unfortunately, these repulsion effects of NF membrane and the enhancing-size method are almost ineffective for congener ions of Sr^2+^ and Mg^2+^ [17]. There is no doubt that it is still difficult for the NF membrane to separate Sr^2+^ and Mg^2+^ ions because of the similarity of their hydrated ion radii (0.428 nm for Mg^2+^, 0.412 nm for Sr^2+^), in addition to their chemical properties [13,17]. 

Recent studies found the separation of congener ions based on the differences in ion hydration free energies [18,19] or electromigration rate [20,21]. However, the chemical structure of the membrane pore needed to be finely controlled in order to preferentially dehydrate a certain ion [18,19]. For the electromigration separation, the selectivity increases with decreasing flow rate. However, the productivity was often low and large-scale application was difficult due to a very low convective velocity employed [20,21]. 

Based on electrokinetic properties of the charged solutes and physicochemical properties of charged NF membrane in the feed solution, an electric field was applied with a cell configuration of anode-NF membrane-cathode, named as electro-NF (ENF), for the improvement the separation performance [22,23]. In our recent work, an electro-NF (ENF) process was employed to separate Mg^2+^ and Li^+^ from the MgCl_2_/LiCl feed solution, achieving an ultrahigh Mg^2+^ rejection due to positive-charge restructuring and enhancement under EF. Li^+^ dehydration was confirmed through molecular dynamics (MD) simulation [24]. This is because the electric field (EF) directly affected the water dipole orientation around the ions [25] and even overcame the hydration free energy of cations to water molecules to dissociate the water molecules bound to the cations in solution [26]. 

In this work, separation of Mg^2+^ and Sr^2+^ through the ENF process is investigated for the first time. In principle, EF can be used to promote preferential dehydration Sr^2+^ ions because of the lower hydration free energy of Sr^2+^ (−1379 kJ/mol) compared with that of Mg^2+^ (−1828 kJ/mol) [13,17], which would promote Sr^2+^ permeation through an NF membrane [24]. Therefore, Sr^2+^/Mg^2+^ separation performance of a ENF process was investigated under different process conditions. Molecular dynamics (MD) simulation can demonstrate the preferential dehydration of Sr^2+^ ions under EF.

## 2. Materials and Methods

### 2.1. Materials

Piperazine (PIP, CAS#110-85-0, Aladdin Reagent Co., Ltd., Shanghai, China) and trimesoyl chloride (TMC, CAS#4422-95-1, Aladdin Reagent Co., Ltd., Shanghai, China), n-hexane (CAS#110-54-3, Tianjin Kemiou Chemical Reagent Co., Ltd., Tianjin, China) were used for interfacial polymerization to prepare polyamide NF membrane. The NF membrane with a pore diameter of 0.940 nm (NF2, in Appendix A) was fabricated from 1 wt.% PIP aqueous solution and 0.25 wt.% TMC/n-hexane solution on PES/SPSf ultrafiltration membrane. Several analytical grade inorganic salts, including MgCl_2_, SrCl_2_, KCl, NaCl, and CaCl_2_ were purchased from Tianjin Kemiou Chemical Reagent Co., Ltd. for the preparation of feed solution. All the solutions in this work were formulated with ultra-pure water with a conductivity of 1.1 μS/cm and a pH of 5.7.

### 2.2. Separation of Mg^2+^ and Sr^2+^ Ions

The separation experiment was carried out with the self-made ENF crossflow device shown as Figure 1. The effective membrane area is 7.07 cm^2^. The diaphragm pump and DC power supply were used to provide pressure and current, respectively. Titanium ruthenium electrodes with porous structures were used as anode and cathode, respectively [24]. 

The separation experiments of Mg^2+^/Sr^2+^ were carried out under a constant current mode at an operation pressure of 5 bar, 25 °C. The current was fixed at 0, 5, 10, 15, or 20 mA, the corresponding current density was 0, 0.71, 1.41, 2.12, or 2.83 mA·cm^−^^2^, respectively. The voltages ranged in 0–3.5 ± 0.1 V. Each membrane was stabilized with de-ionized water for 0.5 h under 6 bar before it was tested on true feed solution. The flow rate of pure water at a pressure of 5 bar was 19.8 mL·s^−^^1^.

The flux (*J*, L·m^−^^2^·h^−^^1^) was calculated from the following equation:(1)$$J=\frac{\Delta V}{A\cdot \Delta t}$$
where Δ*V* (L) is the volume of the permeate, *A* (m^2^) is the effective membrane filtration area, and Δ*t* (h) is the measurement time.

In ENF, most of Mg^2+^ ions could be rejected by the NF membrane. A small amount of Mg^2+^ ions was permeated through the NF membrane, but further precipitated by OH^−^ generated from the cathode reaction described in Figure 2. Therefore, the obtained permeate solution only had very tiny amount of Mg^2+^ ions. In order to clearly describe Mg^2+^ removal by the NF membrane and the whole ENF process, two rejection concepts for Mg^2+^ are proposed. One is membrane rejection of Mg^2+^ (*R*_M_), representing the Mg^2+^ rejection only by an NF membrane; the other is process rejection of Mg^2+^ (*R*_P_), representing the Mg^2+^ rejection by the whole ENF process including membrane rejection and Mg(OH)_2_ precipitate. *R*_p_ was calculated from Mg^2+^ concentrations in the permeate and in the feed according to Equation (2). For *R*_M_, the permeate amount of Mg^2+^ should be plus the amount consumed by precipitation. *R*_M_ was calculated according to Equation (3). The amount of Mg^2+^ in the precipitate was obtained by the following method. The cathode was washed with 0.1 mol/L HCl solution to fully solve Mg(OH)_2_ precipitate. All cleaning liquid was collected to determine the Mg^2+^ concentration in the cleaning liquid through inductively coupled plasma-optical emission spectroscopy (ICP-OES, Agilent, 5100, Santa Clara, CA, USA).

The *R*_P_ and *R*_M_ were calculated according to the following equations:(2)$$R_{P}=\left(1-\frac{C_{\mathrm{Mg},p}}{C_{\mathrm{Mg},f}}\right)\times 100\%$$
(3)$$R_{M}=1-\frac{C_{\mathrm{Mg},P}+n_{\mathrm{Preci}/V_{P}}}{C_{\mathrm{Mg},f}}\times 100\%\;$$
where *C*_Mg,p_ (mol/L), and *C*_Mg,f_ (mol/L) are the concentrations of Mg^2+^ ions in the permeate and in the feed, respectively. *n*_preci_ (mol) represents the amount of Mg^2+^ precipitated on the cathode, *V*_P_ is the volume of permeation. The concentrations for Mg^2+^ and Sr^2+^ were examined by ICP-OES with a determination limit of 0.01 ppm. Each experiment was carried out at least three times, and a mean value was taken.

The separation factor of Sr^2+^ over Mg^2+^ (*S*_Sr,Mg_) was evaluated by the following equation:(4)$$S_{\mathrm{Sr},\mathrm{Mg}}=\frac{C_{\mathrm{Sr},p}/C_{\mathrm{Mg},p}}{C_{\mathrm{Sr},f}/C_{\mathrm{Mg},f}}$$

### 2.3. Molecular Dynamics (MD) Simulation of the Effect of Electric Field on Hydrated Mg^2+^ and Sr^2+^

The simulation was implemented by GROMACS software. First, a box containing one Sr^2+^ ion, two Cl^−^ ions, and 4052 water molecules with 5 nm side length was constructed, with the ion concentration close to infinite dilution (*C*_Sr_ = 1.3711 × 10^−^^8^ mol/L) [25]. Then, energy minimization was firstly performed, and balanced the system capability. Subsequently, a uniform electric field of 0–10 V/nm was applied along the *Z*-axis and run for 3 ns to carry out the simulation of the hydration of ions under the electric field conditions. The hydration of Mg^2+^ ions was also simulated in the same steps.

## 3. Results and Discussion

### 3.1. Effect of Current Density on Rejection of Mg^2+^ or Sr^2+^ in Single Salt System

The permeate flux and rejection performance of MgCl_2_ and SrCl_2_ single salt solutions by ENF process are evaluated in Figure 1. As shown in Figure 1a, the process rejection of Mg^2+^, R_P_, rose from 65% to 94% with increasing current density from 0 to 2.83 mA·cm^−^^2^. The membrane rejection of Mg^2+^, R_M_, also rose. The change was consistent with those in our previous work due to positive-charge restructuring of the NF membrane surface and enhancement under EF [24]. The R_M_ was all lower than the R_p_. The difference between R_M_ and R_p_ is because the partially permeated Mg^2+^ ions reacted with OH^−^ produced on the cathode to form Mg(OH)_2_ precipitate and deposited on the cathode surface, which significantly reduced the content of Mg^2+^ in the permeate. Surprisingly, Sr^2+^ rejection shows a decline from 42% to −80% with the increasing current density from 0 to 2.83 mA·cm^−^^2^, possibly because the EF promoted dehydration of Sr^2+^ ions due to its low hydration free energy to readily passed through membrane pores [24,27,28], which would be discussed in the later. 

The permeate fluxes for MgCl_2_ and SrCl_2_ solutions were decreased with the increase in the current density due to the increase of electroviscous effect with the increasing current density [29,30]. The completely opposite trends of Mg^2+^ and Sr^2+^ rejection provided an opportunity for the separation of MgCl_2_ and SrCl_2_ mixed solution under EF.

### 3.2. Effect of Current Density on Separation of Mg^2+^/Sr^2+^ in Mixed Salt System

When MgCl_2_ and SrCl_2_ aqueous solution were mixed, the interaction among Mg^2+^, Sr^2+^ ions and membrane surface could be different from that in the single salt system. The separation of Mg^2+^/Sr^2+^ under different current densities by the ENF process are shown in Figure 2. Mg^2+^ rejection is about 58% (Figure 2a) when there was no EF applied. When an EF was applied, the R_p_ and R_M_ for Mg^2+^ were all increased firstly and then tended to a stable platform with the increase of current density. At 2.83 mA·cm^−^^2^, R_M_ was 76.1%; and R_p_ was over 99%, higher than that in single salt system because more Mg^2+^ ions were precipitated on the cathode. Meanwhile, Sr^2+^ rejection gradually decreased with the increase of current density, reaching a negative value of −81.4% at 2.83 mA·cm^−^^2^, suggesting that Sr^2+^ permeance was increased with increasing the current density due to the enhancement of Sr^2+^ dehydration under high EF. As can be seen in Figure 2b, the separation factor, S_Sr,Mg_, markedly rose from 1.5 to 180.9 with the current density from 0 to 2.12 mA·cm^−^^2^, respectively, which is 119.6 times higher than that under no EF. The flux was decreased from 71.7 L·m^−^^2^·h^−^^1^ to 23.3 L·m^−^^2^·h^−^^1^ upon increasing the current density from 0 to 2.83 mA·cm^−^^2^ due to the increase of electroviscous effect [29,31].

### 3.3. Effect of Mg^2+^/Sr^2+^ Mass Ratio on Separation of Mg^2+^/Sr^2+^ in Mixed Salt System

Mg^2+^/Sr^2+^ mass ratio is a significant quota in separation performance. The feed solutions with different Mg^2+^/Sr^2+^ mass ratios (fixing Sr^2+^ concentration of 200 ppm) were utilized to evaluate the separation performance of Mg^2+^/Sr^2+^ (Figure 3). Both Mg^2+^ and Sr^2+^ rejections were increased with the increasing Mg^2+^/Sr^2+^ mass ratio without EF in Figure 3a, attributed to the strong screening effect caused by increasing concentration [32,33]. However, when current density was at 2.12 mA·cm^−^^2^, R_p_ (Mg^2+^) stayed above 99% with the growth of Mg^2+^/Sr^2+^ mass ratio, and R_M_ (Mg^2+^) was hardly changed (Figure 3b). While Sr^2+^ rejection rose with the Mg^2+^/Sr^2+^ mass ratio. It is because that more cations gathering near the membrane surface enhanced repulsion between membrane and cations to hold back the permeation of Sr^2+^ ions although easier dehydration of Sr^2+^ ions. The fluxes were all slightly decreased with the increase in Mg^2+^/Sr^2+^ mass ratio due to the increase of osmotic pressure (Figure 3c). Correspondingly, the separation factors for Sr^2+^/Mg^2+^ were decreased, from 928, 180, 80, to 71 at Mg^2+^/Sr^2+^ mass ratios from 0.5/1, 1/1, 2/1 to 3/1, respectively (Figure 3d). It was suggested that low Mg^2+^/Sr^2+^ mass ratio was conducive to the separation of Mg^2+^/Sr^2+^. 

### 3.4. Effect of pH of Feed on Separation of Mg^2+^/Sr^2+^ in Mixed Salt System

The surface charge of NF membrane exhibits strong correlation with pH, affecting rejection performance of the NF membrane [33]. As shown in Figure 4, the effect of pH on Mg^2+^/Sr^2+^ separation was evaluated in the pH range of feed solution from 3 to 8. When the current density was 0 in Figure 4a, Mg^2+^ rejection decreased gradually from 69.4% to 49.2% due to the increase of electronegativity when pH rose from 3 to 8 [34,35], respectively. However, Sr^2+^ rejection firstly reduced from 48.1% to 31% upon increasing pH from 3 to 6, then increased to 47.4% at pH 8, showing a quasi-symmetric rejection curve similar to the symmetric salt system (like NaCl) [11]. In this case, Sr^2+^ ions had a higher permeance (calculated from *J*_Sr_ = *J* × *C*_Sr,P_) of 0.071 mol·m^−^^2^·h^−^^1^ at pH 5–7 than those at acidic or alkaline conditions (permeance of 0.055 and 0.065 mol·m^−^^2^·h^−^^1^ for pH 3 and 8, respectively). These phenomena were also observed by others [36,37]. It was because that membrane charge transition occurs with changing pH [36].

At the current density of 2.12 mA·cm^−^^2^ in Figure 4b, the R_M_ (Mg^2+^) was gradually decreased when pH of the feed rose from 3 to 8 due to the enhancement of negative charge of the membrane surface; while the R_P_ (Mg^2+^) was always maintained over 99%. This is because the cathodic reaction occurred under EF and generated enough OH^−^ ions to precipitate Mg^2+^ ions, resulting in a high and stable R_P_ (Mg^2+^). Sr^2+^ rejection also displayed the same trends as those under no EF. Sr^2+^ rejection firstly reduced from 20% to a negative value −22.8% upon increasing pH from 3 to 6, then increased to −9.8% at pH 8, suggesting that the surface charges still had an effect on Sr^2+^ rejection even under EF. Sr^2+^ rejection was lower than that under no EF because EF prompts the dehydration of Sr^2+^ ions. As a result, in Figure 4d the S_Sr,Mg_ of 180 was obtained at pH 6. The flux displayed a similar trend whether there is an external EF or not in Figure 4c. However, the flux under EF was lower than that under no EF due to the electroviscous effect [29,30]. 

### 3.5. Effect of Other Coexisting Ions on Separation of Mg^2+^/Sr^2+^

There are always many other ions coexisting with Sr^2+^ and Mg^2+^ ions in source liquid, mainly Li^+^, Na^+^, K^+^ and Ca^2+^ ions. It is necessary to explore the separation performance of Mg^2+^/Sr^2+^ in the presence of other coexisting cation ions (Figure 5). It can be seen that the coexisting ions has significant impacts on the permeation of Sr^2+^ ions. When it coexisted with monovalent ions, Li^+^, Na^+^, or K^+^, in the feed solution, R_P_ (Mg^2+^) was still above 99% in Figure 5a–c and Sr^2+^ rejection increased. The rejection for the coexisting monovalent ions all became more negative compared with that of Sr^2+^ ions, suggesting that the coexisting Li^+^, Na^+^, or K^+^ ions weakened the permeability of Sr^2+^ ions due to the completion permeation of Sr^2+^ ions with Li^+^, Na^+^, or K^+^ ions [24]. The fluxes of the feed solution with Li^+^, Na^+^, or K^+^ ions all increased. These phenomena were possibly because the decrease of Sr^2+^ permeation (Sr^2+^ rejection rose) to leave more pore space to allow water permeation due to a smaller radius of Li^+^, Na^+^, or K^+^ ions compared with Sr^2+^ ions (in Appendix A).

In Figure 5d, divalent Ca^2+^ and Sr^2+^ showed almost the same permeation characteristics in the mixed solution due to the nearly hydrated radius, hydration free energy, and diffusion coefficient of Ca^2+^ and Sr^2+^ (Appendix A). The R_P_ (Mg^2+^) was markedly decreased with the increase of Ca^2+^/Sr^2+^ ratio, because Ca^2+^ ions were completed with Mg^2+^ ions to disturb Mg(OH)_2_ precipitate on the cathode surface. The flux in this system was decreased because of ion scaling on the membrane surface [38,39].

### 3.6. The Effect of Electric Field (EF) on Dehydration of Mg^2+^ and Sr^2+^ Ions by Molecular Dynamics (MD) Simulation 

The strength and stability of hydrated ions have an important effect on ion selectivity and mobility [25,40]. MD simulation was conducted to investigate the dehydration of Mg^2+^ and Sr^2+^ ions under different EFs. The radial distribution function (g(r)) as an important parameter in MD simulation represents the distribution probability of other particles in the ‘target particle’ coordinate space [41], which is often used to evaluate the hydration of molecules or ions in solution. The water coordination number (n(r)) is another criterion to represent the structure of hydrated ions, which are computed by the numerical integration of the radial distribution functions curves [25,40].

For a better understanding of the effects of EFs on dehydration of Mg^2+^ and Sr^2+^ ions, ion-oxygen radial distribution functions g(r) of Mg^2+^ and Sr^2+^ under different EFs were analyzed as shown in Figure 6. There were two peaks and two valleys on g(r) curves of Mg-O and Sr-O, which indicated the formation of hydration shells [40]. Obviously, g(r) curves and n(r) curves of Sr-O showed a remarkable decay tendency even under 4 V/nm (Figure 6b), suggesting the weakening of ion hydration. By contrast, for g(r) curves and n(r) curves of Mg-O (Figure 6a) this phenomenon was not found under even 5 v/nm. However, when the EF strength increased by 10 v/nm, g(r) curves and n(r) curves of Mg-O also exhibited a notable declination like those of Sr-O. From these observed results, it can be reasonably inferred the external EF could promote Sr^2+^ dehydration prior to Mg^2+^ for the facile permeation of Sr^2+^ ions mentioned above.

## 4. Conclusions 

In this work, the ENF process was successfully used for separation the congener ions of Mg^2+^ and Sr^2+^. The effects of current densities, Mg^2+^/Sr^2+^ mass ratios, pH of the feed, and coexisting cations on separation performance were investigated. Results showed that Mg^2+^ rejection (R_P_) almost kept above 99%; Sr^2+^ rejection was decreased with increasing EF. A very high S_Sr,Mg_ of 928 was achieved at optimal conditions, far higher than that without EF. The efficient separation of Mg^2+^ and Sr^2+^ ions in the ENF process is mainly because of high-efficiency Mg^2+^ removal resulting from the NF rejection and precipitation of Mg^2+^ ions with OH- generated on the cathode under EF. Sr^2+^ ions were preferentially dehydrated under EF to promote high permeation of Sr^2+^ ions due to the lower hydration energy of Sr^2+^ ions compared with Mg^2+^.

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
