# Peer review of "High-Efficiency Separation of Mg2+/Sr2+ through a NF Membrane under Electric Field"

_membranes, 2021, doi:10.3390/membranes12010057_

Round 1

Reviewer 1 Report

The experimental results presented in this paper will be fruitful for the membrane community. However, the authors should clarify and improve different key points such as :

  • the theoretical background for NF and electro-filtration process
  • the operating conditions for the expérimental work as well ass the molecular dynamics simulation
  • the mass transfert explanations

Comments: 

  • The authors should precise the value of the current, the flow rate, the pH in the "materials and methods" section.
  • The figure as well as scheme numbers should be checked
  • Introduction section : the impact of the electric Field on the solute dehydration should be explained in details
  • The authors should explained why they used a home-made membrane and how they determined its pore size.

Major comments :

  • Concerning the mass transport phenomema, the authors did not considered the electro migration flux of the charged species usually taken into account in electro-filtration processes. This phenomena should be taken before considering the possible dehydration of the solutes.
  • Concerning the variation of the solvant flux with an applied current, the authors should also consider the elecroosmotic flux.

Reviewer 2 Report

Nanofiltration is mainly used for the separation of divalent ions such as Ca2+, Mg2+ etc or molecules in the molecular weight range of 200-5,000 so it would be more appropriate to restudy the section 3.6 with the presence of other diavelent metal cations instead of Li+, Na+, K+. Furthermore, the authors can discuss the selectivity with the obtained results.

The authors should be mentioned the feed solution medium.

There is no info about the applied voltage in any of the  parameters that studied.

In section 3.4, the number of the Figure should be "Figure 3" instead of "Figure 1".

the authors identified the permeability equation in Eq 4 but no numerical value is given regarding on this.

The authors determined the concentration of Mg+2 with ICP-OES but the results do not given. 

Round 2

Reviewer 2 Report

The paper is accepted in present form.